# PROTOTYPE ANTITHESIS FOR BIOLOGICAL FEW-SHOT CLASS-INCREMENTAL LEARNING

**Binghao Liu[1,2]\*, Han Yang[1,2], Fang Wan[3]\*Fei Gu[1,2]**
[1]DAMO Academy, Alibaba Group, Hangzhou, 310023, China
[2]Hupan Lab, Hangzhou, 310023, China
[3]University of Chinese Academy of Sciences, Beijing, 101408, China

## ABSTRACT

Deep learning has become essential in the biological species recognition task. However, a significant challenge is the ability to continuously learn new or mutated species with limited annotated samples. Since species within the same family typically share similar traits, distinguishing between new and existing (old) species during incremental learning often faces the issue of species confusion. This can result in "catastrophic forgetting" of old species and poor learning of new ones. To address this issue, we propose a Prototype Antithesis (PA) method, which leverages the hierarchical structures in biological taxa to reduce confusion between new and old species. PA operates in two steps: Residual Prototype Learning (RPL) and Residual Prototype Mixing (RPM). RPL enables the model to learn unique prototypes for each species alongside residual prototypes representing shared traits within families. RPM generates synthetic samples by blending features of new species with residual prototypes of old species, encouraging the model to focus on species-unique traits and minimize species confusion. By integrating RPL and RPM, the proposed PA method mitigates "catastrophic forgetting" while improving generalization to new species. Extensive experiments on CUB200, PlantVillage, and Tree-of-Life datasets demonstrate that PA significantly reduces inter-species confusion and achieves state-of-the-art performance, highlighting its potential for deep learning in biological data analysis.

## 1 INTRODUCTION

With advances in data science and computing, computer techniques are crucial in fields like biology, offering solutions for species recognition and evolutionary analysis. However, limited data on newly discovered or mutated species poses a challenge, as data annotation requires expertise and is costly. Additionally, similar traits among species within the same family complicate recognition. These issues—data scarcity and species similarity—can cause confusion between new and existing species during continual learning, leading to poor performance on new species and forgetting of old ones.

To overcome this, we introduce the few-shot class-incremental learning paradigm to biological research, allowing models to be trained on existing data and updated with only a few samples from new species. This approach enables continuous learning from new species while preserving knowledge of previously learned ones, addressing data limitations, and improving scalable species recognition. It underscores the vital role of advanced deep learning in advancing biological discovery.

Few-shot class-incremental learning (FSCIL) typically faces two key challenges: "catastrophic forgetting" and "over-fitting". These issues arise when models must learn new classes with few samples and no access to previous data. Early methods address forgetting through knowledge distillation, where the old model's outputs guide the new model. However, when training samples for new classes are limited, the model becomes more prone to over-fitting. Recent approaches Zhang et al. (2021); Zhou et al. (2022) mitigate forgetting by freezing the feature extractor and selectively updating classifier weights, preserving old class performance. However, in biological contexts, distinguishing between new and previously learned species presents an additional challenge due to their

---

\*Correspondence to: Fang Wan (wanfang@ucas.ac.cn), Binghao Liu (liubinghao.lbh@alibaba-inc.com).

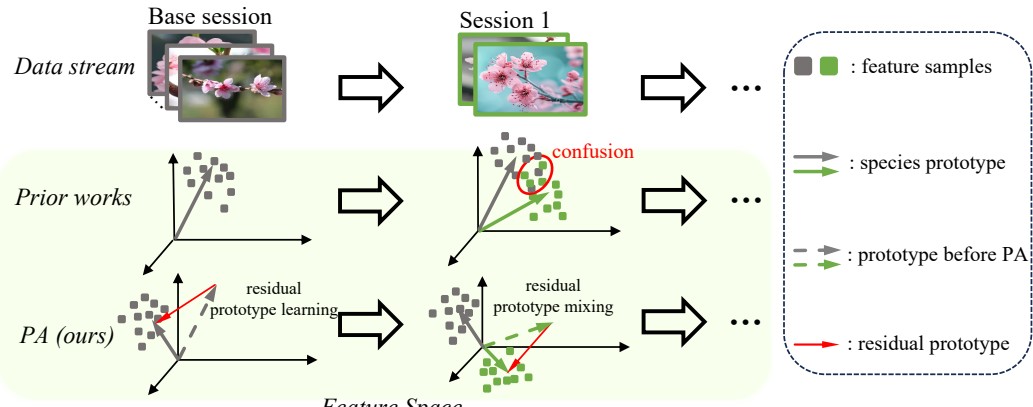

Figure 1: Comparison of prior works and the Prototype Antithesis (PA) method. In incremental learning, when similar new species resemble existing ones, their prototypes tend to be close, leading to confusion during classification (upper). In contrast, our proposed PA addresses this issue by decomposing the prototypes into two parts: the unique prototype specific to a species and the residual prototype shared within its family. The residual prototype is employed in the residual prototype mixing for new species learning, which maximizes the distance between similar species prototypes, thereby reducing classification confusion. (Best viewed in color)

close evolutionary relationships. The similar traits of species within the same family make it difficult for classifiers to differentiate between them. This complicates learning new classes and increases inter-species confusion, reducing performance on previously learned species (Fig. 1, prior works).

In this study, we propose a Prototype Antithesis (PA) method to address these challenges. PA encourages the model to learn both the most discriminative prototypes unique to individual species and the residual prototypes shared by species within the same family. During incremental learning, PA generates new feature samples for model learning by combining residual prototypes from previously learned species with features from new species. This process not only helps the classifier better distinguish between closely related new and old species, but also mitigates over-fitting caused by the limited data available for new species, Fig. 1 (PA).

The PA method operates in two key steps: Residual Prototype Learning (RPL) and Residual Prototype Mixing (RPM). RPL is conducted during base training and can be further divided into two phases. In phase one, images are assigned with "species-level" labels, and those images are used for classification training by calculating the cosine distance between the features and a set of randomly initialized prototypes. After training, those prototypes converge to represent the most discriminative semantics of the corresponding species. In phase two, the species-unique and family-shared semantics are decoupled by a bi-directional optimization process. Specifically, we first compute the residuals between the features and their corresponding prototypes. Then these residual features are assigned with "family-level" labels for training, while the original features are classified by species labels. This bi-directional classification procedure ensures the convergence of species-unique and family-shared traits into their respective prototypes, referred to as species prototypes and family (residual) prototypes. RPM combines the residual prototypes of previously learned species with the new species features using a feature-mixing strategy to generate synthetic feature samples. These synthetic feature samples are labeled as their corresponding new species and are used to facilitate model learning. To classify these synthetic features accurately, the model must capture distinctive traits that differentiate them from residual prototypes and previously learned species, resulting in a well-separated decision boundary. Additionally, the synthetic features act as augmented data, increasing the number of training samples and helping to reduce over-fitting on the new species. Comprehensive experiments conducted on Tree-of-Life Stevens et al. (2024), as well as on the PlantVillage Hughes & Salathé (2015) and CUB200 P. et al. (2010) datasets, demonstrate that our approach achieves state-of-the-art performance.

The contributions of this study are listed below:

- We propose a novel Prototype Antithesis (PA) method to address the challenges of species confusion and over-fitting in few-shot class-incremental learning (FSCIL), particularly within the biological domain.

- We introduce Residual Prototype Learning (RPL) and Residual Prototype Mixing (RPM) to enable the model to capture both species-unique and family-shared traits, focusing on distinguishing between similar new and old species while augmenting training samples.

- Our proposed PA method achieves state-of-the-art performance across both computer vision and biological benchmarks, demonstrating its superiority in addressing the unique challenges of FSCIL.

## 2 RELATED WORKS

**Biological Recognition.** Biological data, characterized by its hierarchical category structure, is frequently used in fine-grained classification tasks, such as those involving the CUB200 dataset P. et al. (2010). These tasks are particularly difficult because subspecies within the same family often display highly similar appearances, making it challenging for models to differentiate them, especially when new species are constantly being discovered. Moreover, these newly identified species often come with limited samples, hindering the model's ability to learn effectively. While recent work like Bio-CLIP Stevens et al. (2024) has introduced foundation models for biological research, it struggles with continual learning and tend to misclassify closely related species.

**Few-shot Learning.** The goal is to learn new classes with limited training samples. Existing methods have made significant progress through metric learning, meta-learning, and data augmentation. Metric learning Vinyals et al. (2016); Snell et al. (2017); Sung et al. (2018); Zhang et al. (2020); Liu et al. (2021); Yang et al. (2021a); Liu et al. (2021); Li et al. (2021) maps new class features into a representation space learned from base classes with sufficient data. Meta-learning Finn et al. (2017); Elsken et al. (2020); Sun et al. (2019) develops optimization strategies to enable models to generalize to new classes with limited data. Data augmentation Zhang et al. (2019); Li et al. (2020); Kim et al. (2020); Yang et al. (2021b) generates synthetic samples to reduce overfitting and support new class learning. However, these methods mainly focus on learning new classes and often overlook the problem of forgetting old classes during incremental learning. GFSL Schonfeld et al. (2019) addresses this by aligning old and new class distributions to mitigate forgetting, but it struggles with large distribution gaps and risks class confusion when old and new classes are highly similar.

**Incremental Learning.** Incremental learning can be divided into task-incremental and class-incremental learning, with the key difference being that task-incremental learning requires task IDs, while class-incremental does not Masana et al. (2020). Existing methods fall into three categories: memory replay, model regularization, and architecture configuration methods. Memory replay Rebuffi et al. (2017); Chaudhry et al. (2018) methods store old class samples to prevent forgetting. Model regularization Li & Hoiem (2018); Dhar et al. (2019) methods add terms to the loss function to reduce forgetting. Architecture configuration Serrà et al. (2018); Mallya & Lazebnik (2018) methods use techniques like model pruning to adjust the feature space. However, these approaches still face over-fitting issues, particularly with limited training samples for new classes.

**Few-shot Class-Incremental Learning.** Under few-shot class-incremental learning (FSCIL) settings, the model is first trained on base classes with sufficient data and then required to generalize to new classes with only a few samples. The primary challenges of FSCIL include not only "catastrophic forgetting" but also the issue of "over-fitting" due to limited training data. Feature topology preservation methods Tao et al. (2020); Zhang et al. (2021) maintain the feature topology of old classes using techniques like neural gas or decoupled training strategies to prevent forgetting. Network regularization methods Akyürek et al. (2021); Kang et al. (2022) align the weight vectors of new classes with those of old classes or selectively update model weights to learn new classes without degrading the performance of old classes. Data generation methods Cheraghian et al. (2021); Liu et al. (2023) generate synthetic feature samples to alleviate over-fitting during incremental learning. However, the confusion between new and old classes remains an unsolved problem in FSCIL. Since the model can only access data from individual incremental sessions, it struggles to learn well-separated decision boundaries between old and new classes, especially when these classes are similar. This confusion can lead to both forgetting of old classes and poor learning of new ones, which is the problem this study aims to address.

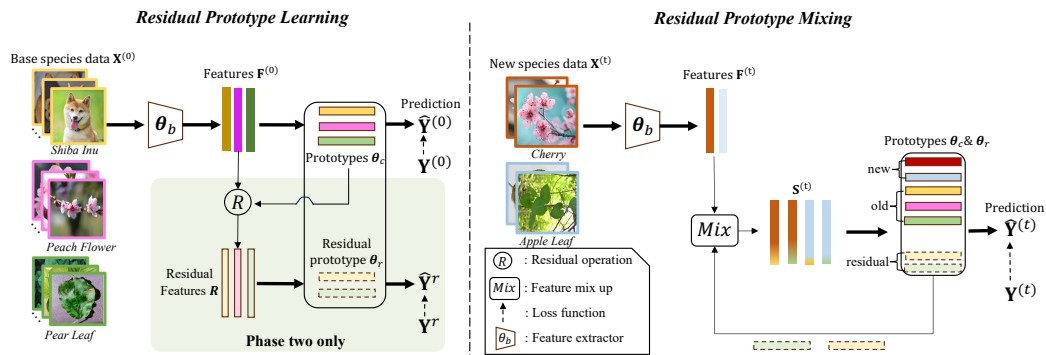

Figure 2: The proposed Prototype Antithesis (PA) method consists of two components: RPL and RPM. RPL has two phases: in phase one, images are encoded by the feature extractor $\boldsymbol{\theta}_b$ and classified using "species-level" labels; in phase two, residuals between the base (old) species features and species prototypes are calculated and classified using "family-level" labels, whereas the original features continue to be classified by "species-level" labels. In RPM, these residual prototypes are mixed with new species features to generate pseudo-samples, optimizing species prototypes and refining the decision boundary. (Best viewed in color)

## 3 PRELIMINARY

Few-shot class-incremental learning can be broadly divided into two stages: the base training stage and the incremental learning stage. During the base training stage, the model is trained using base classes $\mathcal{C}^{(0)}$, which have sufficient training data. In the incremental learning stage, the new classes are divided into $T$ groups, denoted as $\mathcal{C}^{(i)}$, where $i = 1, 2, \ldots, T$, corresponding to the $T$ incremental sessions. In the $i^{\text{th}}$ incremental session, only the dataset $\mathcal{D}^{(i)}$ for classes $\mathcal{C}^{(i)}$ is available for model learning. However, during evaluation, the model is required to classify both old and new classes from the combined set $\{\mathcal{C}^{(0)}, \ldots, \mathcal{C}^{(i)}\}$. Importantly, for any $i_1 \neq i_2$, we have $\mathcal{C}^{(i_1)} \cap \mathcal{C}^{(i_2)} = \varnothing$, meaning that the classes in different incremental sessions are mutually exclusive. This setup imposes the challenge that the model must learn new classes without forgetting previously learned ones, despite not having access to the data from older classes during the incremental sessions.

## 4 METHODOLOGY

In this section, we introduce the proposed PA method. First, we outline the flowchart of the base training and incremental learning procedures. Next, we provide detailed explanations of the residual prototype learning and residual prototype mixing techniques. Finally, we present a theoretical analysis of the proposed prototype antithesis method.

### 4.1 OVERVIEW

The structure of the proposed PA method is illustrated in Fig. 2. The model consists of a feature extractor $\boldsymbol{\theta}_b$, species prototypes $\boldsymbol{\theta}_c = \{\boldsymbol{\mu}_{c_0}, \boldsymbol{\mu}_{c_1}, \boldsymbol{\mu}_{c_2}, \ldots\}$, and residual prototypes $\boldsymbol{\theta}_r = \{\boldsymbol{\mu}_{r_0}, \boldsymbol{\mu}_{r_1}, \boldsymbol{\mu}_{r_2}, \ldots\}$. We employ ResNet-18 as the feature extractor, which encodes images into feature vectors. The prototypes, $\boldsymbol{\theta}_c$ and $\boldsymbol{\theta}_r$, act as classifiers, using cosine distance between the features and prototypes for classification. During base training, the feature extractor and prototypes are optimized using the residual prototype learning strategy. In the incremental learning phase, the species prototypes are further refined through residual prototype mixing, ensuring a well-separated decision boundary between new and existing species. During the inference stage, the features of both old and new species are classified solely by the prototypes, without incorporating the residual.

### 4.2 RESIDUAL PROTOTYPE LEARNING

In standard classification tasks, most existing methods focus on learning the most discriminative representations, allowing classifiers to establish well-separated decision boundaries when all classes

belong to a closed set. However, in few-shot class-incremental learning (FSCIL), where new classes are introduced incrementally, previously learned decision boundaries may struggle to distinguish between new and old classes. For example, when new and old species belong to the same family (e.g., apple leaf and peach leaf), the classifier may confuse them due to their shared "leaf" characteristics.

To address this issue, the model needs to differentiate between "shared" and "unique" traits of species, which is the goal of the proposed Residual Prototype Learning (RPL), Fig. 2 (left). RPL operates in two phases, in phase one, given training samples $\mathbf{X}^{(0)}$ from the base species, the feature extractor $\boldsymbol{\theta}_b$ and species prototypes $\boldsymbol{\theta}_c$ are jointly trained to classify the base species. The training objective for phase one is formulated as follows:

$$
\begin{aligned}
&\underset{\boldsymbol{\theta}_b, \boldsymbol{\theta}_c}{\arg\min} \, \mathcal{L}(\mathbf{Y}^{(0)}, \hat{\mathbf{Y}}^{(0)}; \boldsymbol{\theta}_b, \boldsymbol{\theta}_c) \\
&\text{s.t.} \quad \hat{\mathbf{Y}}^{(0)} = \cos(\mathbf{F}^{(0)}; \boldsymbol{\theta}_c), \quad \mathbf{F}^{(0)} = f(\mathbf{X}^{(0)}; \boldsymbol{\theta}_b),
\end{aligned}
\tag{1}
$$

where $\mathcal{L}$ is the classification loss, $f$ represents the feature extraction process, $\cos$ calculates the cosine similarity between the feature and each prototype individually, resulting in a set of scores, and $\mathbf{Y}^{(0)}$ are the "species-level" labels. After training, the learned species prototypes capture the most discriminative features of the base species. In phase two, for any base species $i$, given the features $\mathbf{F}_i^{(0)}$ and the species prototype $\boldsymbol{\mu}_{c_i}$ (where $\boldsymbol{\mu}_{c_i} \in \boldsymbol{\theta}_c$), the residual features are computed as $\mathbf{R}_i = \mathbf{F}_i^{(0)} - \boldsymbol{\mu}_{c_i}$. Here, $\mathbf{R}_i$ contains both secondary discriminative features and background features, with the secondary discriminative features likely representing traits shared across the family. To decouple "family-level" traits from background noise and the species prototypes (since the most discriminative features might also include some "family-level" characteristics), we assign "family-level" labels to the residual features and classify them by calculating the cosine distance between the residual features and the prototypes. Meanwhile, the original features $\mathbf{F}_i^{(0)}$ are classified based on their "species-level" labels. The optimization objective for base species $i$ in phase two is as:

$$
\begin{aligned}
&\underset{\boldsymbol{\theta}_r, \boldsymbol{\theta}_c}{\arg\min} \left[ \mathcal{L}(\mathbf{Y}_i^r, \hat{\mathbf{Y}}_i^r; \boldsymbol{\theta}_r, \boldsymbol{\theta}_c) + \mathcal{L}(\mathbf{Y}_i^{(0)}, \hat{\mathbf{Y}}_i^{(0)}; \boldsymbol{\theta}_r, \boldsymbol{\theta}_c) \right] \\
&\text{s.t.} \quad \hat{\mathbf{Y}}_i^r = \cos(\mathbf{R}_i; \boldsymbol{\theta}_c, \boldsymbol{\theta}_r), \quad \hat{\mathbf{Y}}_i^{(0)} = \cos(\mathbf{F}_i^{(0)}; \boldsymbol{\theta}_c, \boldsymbol{\theta}_r),
\end{aligned}
\tag{2}
$$

where $\mathbf{Y}_i^r$ and $\mathbf{Y}_i^{(0)}$ denote the "family-level" and "species-level" labels, respectively. The bidirectional optimization process defined in Eq. 2 ensures that the residual prototypes $\boldsymbol{\theta}_r$ converge towards "family-level" traits, while simultaneously driving the species prototypes $\boldsymbol{\theta}_c$ to focus on "species-unique" traits. These prototypes are jointly optimized to refine the decision boundary, effectively distinguishing between traits that are unique to a species and those shared across the family. Please note that if a single species is restricted to a specific environment during base training, environmental features might be misinterpreted as species-specific traits. This issue can be resolved by enhancing the diversity of the data used during the base training phase.

## 4.3 Residual Prototype Mixing

Inspired by human's reference-based learning Kriegeskorte & Douglas (2018), we propose Residual Prototype Mixing (RPM), which drives the model to learn new species by referencing existing knowledge, Fig. 2 (right). The residual prototypes, optimized by Eq. 2, represent family-shared traits likely to appear in closely related species during new species learning. To address this potential confusion, we mix the residual prototypes of old species with new species features to encourage the model to capture the unique characteristics of the new species.

Specifically, in the $t^{\text{th}}$ incremental session, given the residual prototypes $\boldsymbol{\theta}_r$ and features $\mathbf{F}^{(t)}$, the features of the $k^{\text{th}}$ new species are represented as $\mathbf{F}_k^{(t)}$, where this species belongs to the $j^{\text{th}}$ family. We generate synthetic feature samples as $\mathbf{S}_k^{(t)} = \mathbf{F}_k^{(t)} + \boldsymbol{\mu}_{r_j}$, and these synthetic feature samples are assigned their original "species-level" labels for classification, with the optimization process for the $k^{\text{th}}$ species formulated as:

$$
\begin{aligned}
&\underset{\boldsymbol{\theta}_c}{\arg\min} \, \mathcal{L}\left(\mathbf{Y}_k^{(t)}, \hat{\mathbf{Y}}_k^{(t)}; \boldsymbol{\theta}_c, \boldsymbol{\theta}_r\right) \\
&\text{s.t.} \quad \hat{\mathbf{Y}}_k^{(t)} = \cos\left(\mathbf{S}_k^{(t)}; \boldsymbol{\theta}_c, \boldsymbol{\theta}_r\right),
\end{aligned}
\tag{3}
$$

where $\mathbf{Y}_k^{(t)}$ denotes the "species-level" labels. To explicitly explain the optimization process of Eq. 3, we decompose the new species features into species-unique and the other (family-shared and background) features: $\mathbf{F}_k^{(t)} = \mathbf{F}_{ku}^{(t)} + \mathbf{F}_{ko}^{(t)}$, where $\mathbf{F}_{ku}^{(t)}$ and $\mathbf{F}_{ko}^{(t)}$ are orthogonal, meaning they represent non-overlapping features. Consequently, the synthetic features are expressed as: $\mathbf{S}_k^{(t)} = \left(\mathbf{F}_{ko}^{(t)} + \boldsymbol{\mu}_{r_j}\right) + \mathbf{F}_{ku}^{(t)} = \mathbf{F}_{ko}'^{(t)} + \mathbf{F}_{ku}^{(t)}$, where $\mathbf{F}_{ko}'^{(t)} = \mathbf{F}_{ko}^{(t)} + \boldsymbol{\mu}_{r_j}$ represents the other features enhanced with family-shared traits. For the new species prototypes, the limited training data makes it challenging for the model to learn accurate prototypes. Consequently, these prototypes often contain coarse semantic information, combining both the most and second most discriminative features, which are decomposed into family-shared (residual) and species-unique components: $\boldsymbol{\mu}_{c_k} = \boldsymbol{\mu}_{r_j} + \boldsymbol{\mu}_{c_k u}$. Thus, the cosine similarity between $\mathbf{S}_k^{(t)}$, $\boldsymbol{\mu}_{c_k}$, and $\boldsymbol{\mu}_{r_j}$ are calculated as:

$$\cos(\mathbf{S}_k^{(t)}; \boldsymbol{\mu}_{c_k}) = \frac{\left(\mathbf{F}_{ko}'^{(t)} + \mathbf{F}_{ku}^{(t)}\right) \cdot \left(\boldsymbol{\mu}_{r_j} + \boldsymbol{\mu}_{c_k u}\right)}{\left\|\mathbf{S}_k^{(t)}\right\| \cdot \|\boldsymbol{\mu}_{c_k}\|}, \cos(\mathbf{S}_k^{(t)}; \boldsymbol{\mu}_{r_j}) = \frac{\left(\mathbf{F}_{ko}'^{(t)} + \mathbf{F}_{ku}^{(t)}\right) \cdot \boldsymbol{\mu}_{r_j}}{\left\|\mathbf{S}_k^{(t)}\right\| \cdot \|\boldsymbol{\mu}_{r_j}\|}. \quad (4)$$

To avoid misclassifying $\mathbf{S}_k^{(t)}$ as the $j^{\text{th}}$ family label, the difference term of Eq. 4 $\Delta \cos = \cos(\mathbf{S}_k^{(t)}; \boldsymbol{\mu}_{c_k}) - \cos(\mathbf{S}_k^{(t)}; \boldsymbol{\mu}_{r_j})$ must be maximized. As cosine similarity depends only on the angle between vectors, we assume $\boldsymbol{\mu}_{c_k}$ and $\boldsymbol{\mu}_{r_j}$ are unit vectors, and the difference term simplifies to[*]: $\Delta \cos = \frac{\mathbf{F}_{ku}^{(t)} \cdot \boldsymbol{\mu}_{c_k u}}{\left\|\mathbf{S}_k^{(t)}\right\|}$. Since only $\boldsymbol{\mu}_{c_k u}$ is the optimization target, maximizing $\Delta \cos$ equals maximizing $\mathbf{F}_{ku}^{(t)} \cdot \boldsymbol{\mu}_{c_k u}$, which enhances the alignment between the unique features of the new species and its prototype, improving the model's ability to focus on distinctive features. Furthermore, we mix new species features with all previously learned residual prototypes to generate additional synthetic features. The features mixed with unrelated family (residual) prototypes are then used as augmented training data to help reduce model over-fitting.

## 4.4 THEORETICAL ANALYSIS

We analyze Prototype Antithesis (PA) from the perspective of prototype independence and sample concentration. Given prototypes $\boldsymbol{\theta}_c = \{\boldsymbol{\mu}_{c_0}, \boldsymbol{\mu}_{c_1}, \boldsymbol{\mu}_{c_2}, ...\}$ including both old and new species, we define the class separability (CS) as:

$$\text{CS} = \frac{\|\boldsymbol{\mu}_{c_i} - \boldsymbol{\mu}_{c_j}\|_2^2}{\sum_{m=1}^{N_i}(1 - \cos(\mathbf{f}_m; \boldsymbol{\mu}_{c_i})) + \sum_{v=1}^{N_j}(1 - \cos(\mathbf{f}_v; \boldsymbol{\mu}_{c_j}))}, \quad (5)$$

where $i$ and $j$ denote any two different species, while $\mathbf{f}_m$ and $\mathbf{f}_v$ represent their feature samples. In this formulation, the numerator measures the independence or separation of the species prototypes $\boldsymbol{\mu}_{c_i}$ and $\boldsymbol{\mu}_{c_j}$ (unit vectors), while the denominator reflects the concentration of samples around their respective species prototype, quantifying how well the samples align with their class center. The goal of a classification model is to maximize class separability, where the large separation between species prototypes and tight concentration of samples around their respective prototypes leads to better classification performance. Without loss of generality, let us assume that the prototypes of an old species $\boldsymbol{\mu}_{c_i}$ and a new species $\boldsymbol{\mu}_{c_j}$ are related as $\boldsymbol{\mu}_{c_j} = \hat{\boldsymbol{\mu}}_{c_i} + \delta, \|\hat{\boldsymbol{\mu}}_{c_i}\| \in [0, 1]$, where $\hat{\boldsymbol{\mu}}_{c_i}$ represents the similar parts with $\boldsymbol{\mu}_{c_i}$, and $\delta$ is orthogonal to $\boldsymbol{\mu}_{c_i}$ and $\hat{\boldsymbol{\mu}}_{c_i}$, representing the unique parts. Substituting this into Equation 5, we can rewrite the class separability as[†]:

$$\text{CS} = \frac{1 + \delta^2 - \hat{\boldsymbol{\mu}}_{c_i} \cdot (2 - \hat{\boldsymbol{\mu}}_{c_i})}{\sum_{m=1}^{N_i}(1 - \cos(\mathbf{f}_m; \boldsymbol{\mu}_{c_i})) + \sum_{v=1}^{N_j}(1 - \cos(\mathbf{f}_v; \boldsymbol{\mu}_{c_j}))}. \quad (6)$$

While all methods minimize the denominator using classification loss, the key difference lies in the numerator. The term $\hat{\boldsymbol{\mu}}_{c_i}(2 - \hat{\boldsymbol{\mu}}_{c_i})$ is monotonically increasing over the interval $[0, 1]$. In conventional approaches, when the new species prototype $\boldsymbol{\mu}_{c_j}$ is similar to the old one $\boldsymbol{\mu}_{c_i}$, $\hat{\boldsymbol{\mu}}_{c_i}$ becomes large, indicating prototype overlap and reducing the numerator in Eq. 6. However, PA reduces this overlap by emphasizing species-unique features ($\delta$), decreasing $\hat{\boldsymbol{\mu}}_{c_i}$, thereby increasing the numerator in Eq. 6 and improving class separability.

---

[*]The proof is included in the "Appendix".

[†]Please refer to the "Appendix" for the detailed derivation.

Table 1: Ablation experiments of the proposed "RPL" and "RPM". The baseline method is a standard classification network consisting of a feature extractor and a classifier. "$Mean_{old}$" and "$Mean_{new}$" denote the average accuracy of old and new species across all incremental sessions.

| Baseline | RPL | RPM | Accuracy in each session (%) | | | | | $Mean_{old}$ | $Mean_{new}$ | PD $\downarrow$ |
|---|---|---|---|---|---|---|---|---|---|---|
| | | | 0 | 1 | 2 | 3 | 4 | | | |
| ✓ | | | 99.47 | 93.83 | 90.76 | 90.47 | 86.21 | 95.63 | 81.58 | 13.26 |
| ✓ | ✓ | | 99.47 | 94.21 | 93.56 | 92.12 | 88.86 | 96.89 | 84.17 | 10.61 |
| ✓ | ✓ | ✓ | 99.50 | 95.83 | 93.45 | 93.24 | 90.51 | 98.47 | 86.59 | 8.99 |

## 5 EXPERIMENTS

We conduct comprehensive experiments on several public datasets for model analysis, and compare our proposed method with the state-of-the-art methods under FSCIL settings.

### 5.1 EXPERIMENT SETTINGS

**Datasets.** We adopt CUB200 P. et al. (2010), PlantVillage Hughes & Salathé (2015), and Tree-of-Life Stevens et al. (2024) as benchmarks. In the FSCIL task, classes are split into base and incremental categories. Under the $N$-way $K$-shot setting, the incremental classes are divided into $T$ groups, corresponding to the $T$ incremental sessions. In each session, $N$ classes are presented for training, with each class containing $K$ samples. For the CUB200 dataset, 100 classes are used as base classes, and the remaining 100 classes are divided into 10 sessions ($T = 10$), with each session containing 10 classes (10-way) and 5 samples per class (5-shot). In the PlantVillage dataset, 19 classes serve as base classes, while the remaining 20 classes form incremental classes under a 5-way 5-shot setting. For the Tree-of-Life dataset, we randomly sample 100 classes for evaluation, where 60 are base classes, and the remaining 40 are split into groups under the 5-way 5-shot setting.

**Model Training.** To maintain consistency with prior work, we use a ResNet-18 pretrained on ImageNet as the feature extractor for the CUB-200 dataset, while for the PlantVillage and Tree-of-Life datasets, we also use ResNet-18 as the backbone. The optimizer is SGD, with a "milestone" learning rate decay strategy. For data pre-processing, we apply image normalization, random cropping, random resizing, and horizontal flipping for data augmentation. During base training, we use a batch size of 128 and an initial learning rate of 0.004, training for 100 epochs. In the incremental learning phase, the classifier is optimized over 100 epochs with a learning rate of 0.01. We conducted 5 experiments with different seeds on the fixed datasets and reported the averaged results to minimize randomness. All experiments were performed using Pytorch 1.11.0 on Nvidia A800 GPUs.

**Evaluation.** We adopt "classification accuracy" and "performance drop (PD)" as the evaluation metrics. "Classification accuracy" validates the discriminating ability of the model, and it is defined as $\text{accuracy} = \frac{\text{TP+TN}}{\text{TP+TN+FP+FN}}$, where TP, TN, FP and FN denote the numbers of true positives, true negatives, false positives, and false negatives, respectively. "PD" measures the model forgetting and is defined as $\text{PD} = \text{accuracy}^{(0)} - \text{accuracy}^{(T)}$, where $\text{accuracy}^{(0)}$ and $\text{accuracy}^{(T)}$ denote the model performance in the 0-th and $T$-th sessions, respectively.

### 5.2 ABLATION STUDY

To rigorously assess the effectiveness of our proposed approach, we perform ablation experiments on the PlantVillage dataset.

**Residual Prototype Learning.** As seen in Table 1, the proposed RPL increases performance on old species by 1.26% and on new species by 2.59% over the baseline. This is attributed to the model's ability to classify both unique and shared features of the base species, resulting in a more compact decision boundary that supports learning in both old and new species.

Table 2: Comparison of "CAM", "Grad-CAM" and the proposed "RPL".

| Method | $Mean_{old}$ | $Mean_{new}$ | $Similarity$ |
|---|---|---|---|
| CAM | 95.31% | 81.25% | 6.53% |
| Grad-CAM | 95.32% | 79.82% | 8.15% |
| RPL | **98.47%** | **86.59%** | **1.89%** |

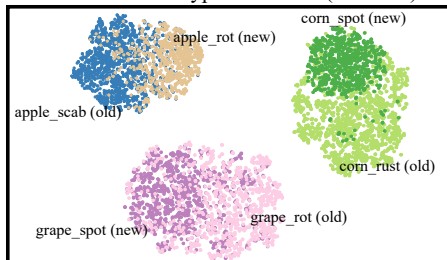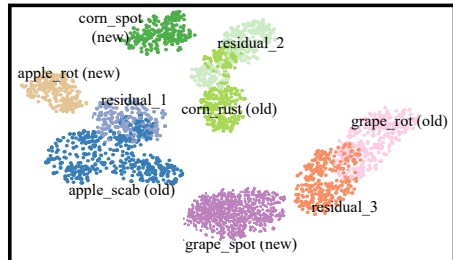

Figure 3: t-SNE visualization of feature distributions comparing models with and without Prototype Antithesis (PA). Without PA, the features of new species overlap with those of similar old species. In contrast, with PA, the new species features are pushed away from the residual features, effectively distinguishing the new and old species and creating a well-separated decision boundary. (Best viewed in color)

**Residual Prototype Mixing.** Table 1 demonstrates that combining RPL with RPM boosts old and new species performance by 2.84% and 5.01%, respectively, and decreases "PD" by 4.37%. RPM reduces the overlap of shared components between new and old species, effectively minimizing species confusion. Together, RPL and RPM work synergistically to reduce "old species forgetting" while enhancing "new species generalization".

**Feature decomposition strategy.** To assess the effectiveness of different feature decomposition strategies, we use CAM Zhou et al. (2016) and Grad-CAM Selvaraju et al. (2017) to distinguish between unique and common features. Specifically, we set a threshold to extract the highest-response regions from the feature maps as unique features, while the secondary-high-response regions represent common features. As shown in Table 2, RPL achieves the highest mean accuracy and the lowest similarity between unique and common features, which can be attributed to its bi-directional optimization that more effectively decouples unique features from shared ones.

## 5.3 MODEL ANALYSIS

In this section, we further conduct comprehensive experiments for detailed statistical results and visualization.

**Feature distribution.** We employ t-SNE to visualize the feature distributions of similar old and new species, as well as the residual feature distributions. As depicted in Fig.3(left), without the proposed prototype antithesis, feature samples of the old and new species overlap at the edges of the distribution. However, with the introduction of the prototype antithesis, the distance between the new species and the residual features is maximized, making the features of the new and old species more distinguishable (Fig.3(right)). This adjustment optimizes the decision boundary and reduces species confusion.

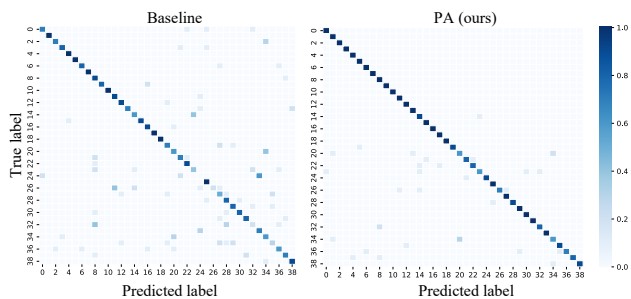

Figure 4: Confusion matrix of the baseline method and our proposed PA approach. Compared with the baseline method, PA significantly reduces the false prediction among old and new species, alleviating species confusion.

**Confusion Matrix.** We plot the confusion matrix for both the baseline method and our proposed PA method. As shown in Fig. 4, the PA method significantly reduces false classifications between old and new species. These results demonstrate that PA effectively minimizes confusion between new and old species, mitigating forgetting of old species while enhancing learning for new species.

Table 3: Performance comparison on the CUB200 dataset. "*" indicates results that were re-implemented using the official code.

| Method | Accuracy in each session (%) | | | | | | | | | | | PD ↓ |
|---|---|---|---|---|---|---|---|---|---|---|---|---|
| | 0 | 1 | 2 | 3 | 4 | 5 | 6 | 7 | 8 | 9 | 10 | |
| TOPIC Tao et al. (2020) | 68.68 | 62.49 | 54.81 | 49.99 | 45.25 | 41.40 | 38.35 | 35.36 | 32.22 | 28.31 | 26.28 | 42.40 |
| SPPR Zhu et al. (2021) | 68.68 | 61.85 | 57.43 | 52.68 | 50.19 | 46.88 | 44.65 | 43.07 | 40.17 | 39.63 | 37.33 | 31.35 |
| SFMS Cheraghian et al. (2021) | 68.78 | 59.37 | 59.32 | 54.96 | 52.58 | 49.81 | 48.09 | 46.32 | 44.33 | 43.43 | 43.23 | 25.55 |
| FSLL Mazumder et al. (2021) | 72.77 | 69.33 | 65.51 | 62.66 | 61.10 | 58.65 | 57.78 | 57.26 | 55.59 | 55.39 | 54.21 | **17.38** |
| CEC Zhang et al. (2021) | 75.85 | 71.94 | 68.50 | 63.50 | 62.43 | 58.27 | 57.73 | 55.81 | 54.83 | 53.52 | 52.28 | 23.57 |
| Meta-FSCIL Chi et al. (2022) | 75.90 | 72.41 | 68.78 | 64.78 | 62.96 | 59.99 | 58.30 | 56.85 | 54.78 | 53.82 | 52.64 | 23.26 |
| FACT Zhou et al. (2022) | 75.90 | 73.23 | 70.84 | 66.13 | 65.56 | 62.15 | 61.74 | 59.83 | 58.41 | 57.89 | 56.94 | 18.96 |
| SOFTNet Kang et al. (2023) | 78.07 | 74.58 | 71.37 | 67.54 | 65.37 | 62.60 | 61.07 | 59.37 | 57.53 | 57.21 | 56.75 | 21.32 |
| WaRP Kim et al. (2023) | 77.74 | 74.15 | 70.82 | 66.90 | 65.01 | 62.64 | 61.40 | 59.86 | 57.95 | 57.77 | 57.01 | 20.73 |
| BiDist Zhao et al. (2023) | **79.12** | 74.99 | 70.87 | 67.30 | 65.89 | 63.45 | 61.40 | 60.11 | 58.61 | 58.23 | 57.48 | 21.64 |
| NC-FSCIL Yang et al. (2023) | 80.45 | 75.98 | 72.30 | **70.28** | 68.17 | 65.16 | 64.43 | 63.25 | 60.66 | 60.01 | 59.44 | 21.01 |
| OrCo* Ahmed et al. (2024) | 74.72 | 66.06 | 64.72 | 62.88 | 61.89 | 59.66 | 58.95 | 59.03 | 57.24 | 57.79 | 57.18 | 17.54 |
| TEEN Wang et al. (2024) | 77.26 | **76.13** | **72.81** | 68.16 | 67.77 | 64.40 | 63.25 | 62.29 | 61.19 | 60.32 | 59.31 | 18.13 |
| **PA(ours)** | 78.69 | 75.59 | 72.71 | 68.71 | **68.37** | **65.77** | **64.75** | **63.59** | **62.76** | **62.02** | **61.19** | 17.50 |

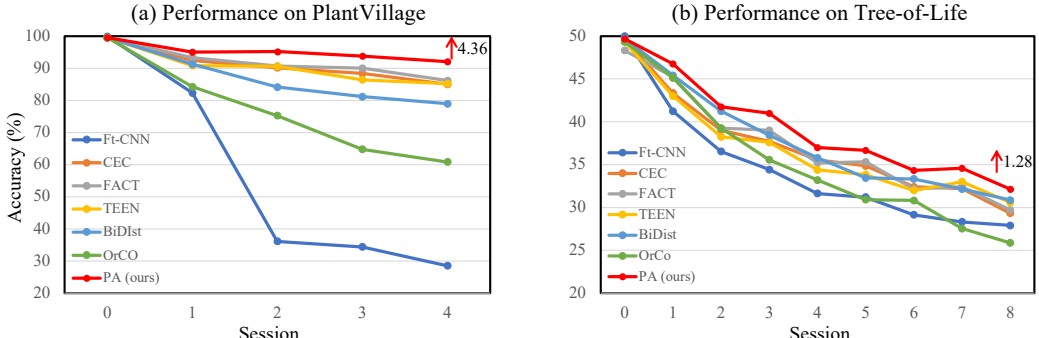

Figure 5: Performance on the PlantVillage and Tree-of-Life datasets.

## 5.4 COMPARISON WITH THE STATE-OF-THE-ART METHODS

We compare PA against state-of-the-art methods on both computer vision benchmarks (e.g., CUB200) and biological datasets (e.g., PlantVillage and Tree-of-Life).

**CUB200:** As shown in Table 3, our proposed PA outperforms the TEEN and OrCo by 1.88% and 4.01%, as well as achieving a lower "PD" compared to most other methods. **PlantVillage:** From Fig. 5(a) we can see that PA outperforms other methods with significant margins. The categories in the PlantVillage dataset (different leaves with disease) share many common traits, and the significant improvement in performance shows PA's superiority in distinguishing closely related old and new species. **Tree-of-Life:** PA outperforms OrCo Ahmed et al. (2024) and BiDist Zhao et al. (2023) by 6.26% and 1.28% on this challenging dataset, respectively, Fig. 5(b). This result confirms PA's robustness in handling high-diversity biological data.

## 6 CONCLUSION

In this paper, we propose a Prototype Antithesis (PA) method to address the issue of species confusion in biological few-shot class-incremental learning. The PA method comprises Residual Prototype Learning (RPL) and Residual Prototype Mixing (RPM). RPL decomposes species-unique and family-shared (residual) prototypes, while RPM generates synthetic samples by blending new species features with residual prototypes. Together, RPL and RPM optimize the decision boundary between old and new species, mitigating "old species forgetting" and enhancing "new species learning". PA offers a novel perspective for recognizing biological species and traits using few-shot class-incremental learning.

AUTHOR CONTRIBUTIONS

Fei Gu served as the project manager. Binghao Liu proposed the idea of the paper, designed the methodology, conducted the experiments, and wrote the manuscript. Han Yang participated in discussions on experimental settings. Fang Wan participated in idea discussions and provided suggestions for figure visualization.

ACKNOWLEDGMENTS

This work was financially supported in part by grants from the Biological Breeding - Major Projects (2023ZD04076).

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

## A APPENDIX

### A.1 DERIVATION OF $\Delta \cos$.

The $\Delta \cos$ in section 4.3 is calculated as below:

$$
\begin{aligned}
\Delta \cos &= \cos(\mathbf{S}_k^{(t)}, \mu_{c_k}) - \cos(\mathbf{S}_k^{(t)}, \mu_{r_j}) \\
&= \frac{\left(\mathbf{F}_{ko}^{\prime(t)} + \mathbf{F}_{ku}^{(t)}\right) \cdot \left(\mu_{r_j} + \mu_{c_k u}\right)}{\left\|\mathbf{S}_k^{(t)}\right\| \cdot \|\mu_{c_k}\|} - \frac{\left(\mathbf{F}_{ko}^{\prime(t)} + \mathbf{F}_{ku}^{(t)}\right) \cdot \mu_{r_j}}{\left\|\mathbf{S}_k^{(t)}\right\| \cdot \|\mu_{r_j}\|} \\
&= \frac{\mathbf{F}_{ko}^{\prime(t)} \cdot \mu_{r_j} + \mathbf{F}_{ko}^{\prime(t)} \cdot \mu_{c_k u} + \mathbf{F}_{ku}^{(t)} \cdot \mu_{r_j} + \mathbf{F}_{ku}^{(t)} \cdot \mu_{c_k u} - \mathbf{F}_{ko}^{\prime(t)} \cdot \mu_{r_j} - \mathbf{F}_{ku}^{(t)} \cdot \mu_{r_j}}{\left\|\mathbf{S}_k^{(t)}\right\| \cdot \|\mu_{r_j}\|} \quad (7) \\
&= \frac{\mathbf{F}_{ko}^{\prime(t)} \cdot \mu_{c_k u} + \mathbf{F}_{ku}^{(t)} \cdot \mu_{c_k u}}{\left\|\mathbf{S}_k^{(t)}\right\| \cdot \|\mu_{r_j}\|}.
\end{aligned}
$$

Since $\mathbf{F}_{ko}^{\prime(t)}$ are orthogonal to $\mu_{c_k u}$ (there are no overlap between them), $\mathbf{F}_{ko}^{\prime(t)} \cdot \mu_{c_k u}$ is equal to 0, and $\mu_{r_j}$ is unit vector, thus Eq. 7 can be rewrite as:

$$
\begin{aligned}
\Delta \cos &= \cos(\mathbf{S}_k^{(t)}, \mu_{c_k}) - \cos(\mathbf{S}_k^{(t)}, \mu_{r_j}) \\
&= \frac{\mathbf{F}_{ku}^{(t)} \cdot \mu_{c_k u}}{\left\|\mathbf{S}_k^{(t)}\right\|}.
\end{aligned}
\quad (8)
$$

Table 4: Performance comparison on the PlantVillage dataset. "*" indicates results that were re-implemented using the official code.

| Method | Accuracy in each session (%) | | | | | PD ↓ |
|---|---|---|---|---|---|---|
| | 0 | 1 | 2 | 3 | 4 | |
| Ft-CNN | **99.89** | 82.33 | 36.11 | 34.38 | 28.53 | 71.47 |
| CEC* | 99.53 | 92.46 | 90.13 | 88.35 | 85.01 | 14.52 |
| FACT* | 99.47 | 93.33 | 90.69 | 90.00 | 86.15 | 13.32 |
| TEEN* | 99.50 | 90.83 | 90.56 | 86.38 | 85.19 | 14.31 |
| BiDist* | 99.51 | 91.25 | 84.14 | 81.18 | 78.97 | 20.54 |
| NC-FSCIL* | 99.48 | **96.05** | **93.59** | 92.88 | 89.13 | 10.35 |
| OrCo* | 99.58 | 84.25 | 75.24 | 64.77 | 60.82 | 38.76 |
| **PA (ours)** | 99.50 | 95.83 | 93.45 | **93.24** | **90.51** | **8.99** |

This is consistent with the result presented in Section 4.3.

## A.2 DERIVATION OF EQ. 6 IN THE PAPER.

The class separability in section 4.4 is written as:

$$
\begin{aligned}
\text{CS} &= \frac{\|\mu_{c_i} - (\hat{\mu}_{c_i} + \delta)\|_2^2}{\sum_{m=1}^{N_i}(1 - \cos(\mathbf{f}_m; \mu_{c_i})) + \sum_{v=1}^{N_j}(1 - \cos(\mathbf{f}_v; \mu_{c_j}))}, \\
&= \frac{\mu_{c_i}^2 + \hat{\mu}_{c_i}^2 + \delta^2 + 2 \cdot \delta \cdot \hat{\mu}_{c_i} - 2 \cdot \mu_{c_i} \cdot (\hat{\mu}_{c_i} + \delta)}{\sum_{m=1}^{N_i}(1 - \cos(\mathbf{f}_m; \mu_{c_i})) + \sum_{v=1}^{N_j}(1 - \cos(\mathbf{f}_v; \mu_{c_j}))}.
\end{aligned}
\tag{9}
$$

Since $\delta$ is orthogonal to $\boldsymbol{\mu}_{c_i}$ and $\hat{\boldsymbol{\mu}}_{c_i}$, and $\mu_{c_i}$ is unit vector, Eq. 10 is rewritten as:

$$
\begin{aligned}
\text{CS} &= \frac{1 + \hat{\mu}_{c_i}^2 + \delta^2 - 2 \cdot \mu_{c_i} \cdot \hat{\mu}_{c_i}}{\sum_{m=1}^{N_i}(1 - \cos(\mathbf{f}_m; \mu_{c_i})) + \sum_{v=1}^{N_j}(1 - \cos(\mathbf{f}_v; \mu_{c_j}))}, \\
&= \frac{1 + \delta^2 - \hat{\mu}_{c_i} \cdot (2 - \hat{\mu}_{c_i})}{\sum_{m=1}^{N_i}(1 - \cos(\mathbf{f}_m; \mu_{c_i})) + \sum_{v=1}^{N_j}(1 - \cos(\mathbf{f}_v; \mu_{c_j}))},
\end{aligned}
\tag{10}
$$

which aligns with Eq. 6 in the paper.

## A.3 PERFORMANCE ON THE PLANTVILLAGE DATASET

We present the detailed classification accuracy on the PlantVillage dataset in Table 4. The results show that our method significantly outperforms other methods in the final session and achieves the smallest performance drop.

## A.4 PERFORMANCE ON THE TREE-OF-LIFE DATASET

The detailed results on the Tree-of-Life dataset are shown in Table 5. Our proposed PA method outperforms most methods while delivering comparable performance (slightly lower) than NC-FSCIL. Unlike NC-FSCIL, however, our method does not require prior knowledge of the total number of unknown classes, making it more scalable and practical.

## B DATASETS STATISTICS

**CUB200.** The CUB-200-2011 dataset contains 11,788 images of birds, categorized into 200 fine-grained subclasses (species) and 37 broader classes (families). **PlantVillage.** The PlantVillage dataset contains 54,306 images of plant leaves, categorized into 38 fine-grained subclasses (species) and 14 broader classes (families). The images include healthy and diseased leaves, making it a widely used benchmark for plant disease classification and recognition tasks. **Tree-of-Life.** We randomly sampled 13,487 images from 100 species in the Tree-of-Life dataset, spanning 42 families.

Table 5: Performance comparison on the Tree-of-Life dataset. "*" indicates results that were re-implemented using the official code.

| Method | Accuracy in each session (%) | | | | | | | | | PD ↓ |
|--------|-------|-------|-------|-------|-------|-------|-------|-------|-------|------|
| | 0 | 1 | 2 | 3 | 4 | 5 | 6 | 7 | 8 | |
| Ft-CNN | 49.98 | 41.25 | 36.53 | 34.42 | 31.64 | 31.20 | 29.15 | 28.32 | 27.91 | 22.07 |
| CEC* | 49.68 | 43.35 | 39.02 | 37.71 | 35.56 | 34.87 | 32.41 | 32.21 | 29.33 | 20.35 |
| FACT* | 48.33 | 45.25 | 39.25 | 39.00 | 35.20 | 35.33 | 32.17 | 32.43 | 29.71 | 18.62 |
| TEEN* | 49.33 | 43.00 | 38.25 | 37.60 | 34.40 | 33.83 | 32.00 | 33.00 | 30.57 | 18.76 |
| BiDist* | 49.67 | 45.43 | 41.25 | 38.44 | 35.80 | 33.45 | 33.33 | 32.15 | 30.86 | 18.81 |
| NC-FSCIL* | **49.89** | 46.52 | **42.08** | **41.23** | 37.56 | 36.49 | **34.45** | 34.21 | **32.42** | **17.47** |
| OrCo* | 49.33 | 45.14 | 39.25 | 35.56 | 33.20 | 30.91 | 30.83 | 27.54 | 25.86 | 23.47 |
| **PA (ours)** | 49.67 | **46.75** | 41.75 | 41.00 | **37.68** | **36.67** | 34.33 | **34.57** | 32.14 | 17.53 |

This dataset exhibits high diversity, making it a challenging benchmark for the few-shot incremental learning task.

