# OpenReview forum: "Prototype antithesis for biological few-shot class-incremental learning"
_ICLR.cc/2025/Conference — ICLR 2025 Poster_

### Official Review · Reviewer_Zao3 · 2024-10-19

**Soundness:** 3
**Presentation:** 3
**Contribution:** 3
**Rating:** 6
**Confidence:** 3

**Summary:**

The article proposes a Prototype Antithesis (PA) method, which first learns species-unique and family-shared semantics of original classes. For new classes, it promotes learning by mixing family-shared features while preserving the model's ability to discriminate original classes. Experimental results validate the effectiveness of this method.

**Strengths:**

1. The paper is well-organized.
2. The literature review is detailed and comprehensive.
3. The suite of experiments is admittedly comprehensive.

**Weaknesses:**

1. There are three variable in cosin similarity calculation, please explain how the cosine similarity is specifically calculated in Eq 2 for  $\hat{Y}_{i}^r=cos(R_i, \theta_c, \theta_r)$ and $\hat{Y}_i^0 = cos(F_i^0; \theta_c, \theta_r)$.
2. How is the family-level label defined? Is this label provided by the dataset?
3. What is the number of features for $\theta_r$, and does this number change during training?
4. Why does mixing the residual prototypes of old species with new species features encourage the model to capture the unique characteristics of the new species?
5. Does the method require integrating residual features during the inference phase?

**Questions:**

The technical details are unclear; please see the weaknesses section.

---

> ### Author Response · Authors · 2024-11-19
> **The responses to Reviewer Zao3**
>
> We sincerely appreciate your valuable review comments and the effort you have put into them. We have addressed each of the questions you raised in detail. If anything is unclear, please inform us, and we will respond promptly.
>
> >*W1: Questions about cosine similarity calculation in Eq. 2.*
>
> **Response:** The cosine similarity is computed by evaluating the similarity between the feature $F$ and each prototype individually, resulting in a set of scores that serves as the standard output of the classification model. Specifically, for the term $\cos\left( F; \boldsymbol{\theta}_c, \boldsymbol{\theta}_r \right)$, we calculate the similarity score between $F$ and each prototype in $\boldsymbol{\theta}_c \cup \boldsymbol{\theta}_r$. These similarity scores collectively form the output of the model. To ensure clarity, we have revised the term from $ \cos\left( F, \boldsymbol{\theta}_c, \boldsymbol{\theta}_r \right) $ to $ \cos\left( F; \boldsymbol{\theta}_c, \boldsymbol{\theta}_r \right) $ and further explained this calculation in Section 4.2 of the revised paper.
>
> >*W2: Is this label provided by the dataset?*
>
> **Response:** Yes, the family-level labels are obtained directly from the dataset. For example, in CUB-200, species such as "Black-footed Albatross," "Laysan Albatross," and "Sooty Albatross" are assigned the family-level label "Albatross".
>
> >*W3: What is the number of features for $\boldsymbol{\theta}_r$, and does this number change during training?*
>
> **Response:** The number of features for $\boldsymbol{\theta}_r$ is equal to the number of families of base species, and it is fixed during training.
>
> >*W4: How RPM encourage the model to capture the unique characteristics of the new species?*
>
> **Response:** As described in Lines 256–296, this paragraph explains how RPM helps the model focus on "species-unique" features in new categories based on the classification model's optimization process. To clarify, both features and prototypes are decomposed into species-related and species-unrelated components. For features, the species-unrelated part is further divided into family-shared and background components, while for prototypes, the species-unrelated part consists solely of the family-shared component.
>
> When the new and old species belong to the same family, RPM enhances the family-shared part of the new species by mixing the old species' family-shared features with the new species' features. This makes the classifier likely to produce a high response to the family-shared part, potentially causing the model to misclassify it as the family label. To avoid this, the model enhances the response of the species-unique components, which is achieved through the optimization of Eq. 3, as mathematically detailed in Lines 285–291 of the original paper.
>
>
> >*W5: Does the method require integrating residual features during the inference phase?*
>
> **Response:** No, during the inference stage, the features of new species are directly classified by the prototypes without incorporating residual features. This is because, through RPM, the model has established well-separated decision boundaries, allowing it to achieve high accuracy when directly classifying new categories. We have included this clarification in the revised paper (Section 4.1).

---

> > ### Comment · Reviewer_Zao3 · 2024-11-24
> >
> > Thank you for your reply. All my concerns are addressed, I will raise my vote to 6.

---

> > > ### Author Response · Authors · 2024-11-24
> > > **The responses to Reviewer Zao3**
> > >
> > > Thanks for your recognition of our work and the valuable feedback you have provided, which has greatly helped us improve the quality of the manuscript!

---

### Official Review · Reviewer_KRan · 2024-11-04

**Soundness:** 3
**Presentation:** 2
**Contribution:** 2
**Rating:** 8
**Confidence:** 3

**Summary:**

The authors present a novel approach to handle the few-shot class-incremental learning problem, specifically in species classification. The authors take advantage of the hierarchical taxonomic relationship between species and use their family label to transfer knowledge from existing species and new species. Their approach, named Prototype Antithesis (PA), employs Residual Prototype Learning (RPL) to learn unique prototypes per species and novel residual prototypes to represent shared traits between species in a family. To further improve the model’s discriminating ability, they utilize Residual Prototype Mixing to generate synthetic samples for data augmented training. They show SOTA performance on three biological datasets: CUB200, PlantVillage, and Tree-of-Life.

**Strengths:**

- They show the generalization of their ability by evaluating on three biological image datasets.
- The authors present novel residual prototype learning and residual prototype mixing training methods that improve class separating and alleviate catastrophic forgetting.
- They present a theoretical analysis of their method, focusing on class-separation.
- An ablation study shows the quantitative effectiveness of their residual prototype learning and mixing methods.
- Their method shows stronger performance than others in the later stages of the incremental learning.

**Weaknesses:**

- The feature decomposition strategy section is convoluted, so Table 2 doesn’t make sense to me. I understand that there are high response and secondary response regions being extracted, but I’m confused how they are used to obtain the values in Table 2.
- You have all of your model’s accuracies bolded in Table 3, but at several columns, your model is not the best (columns 0, 1, 2, and the PD). Please correct.
- The method is only evaluated on 100 classes in the tree-of-life dataset. This should be run multiple times to ensure the robustness across different sections of the dataset given its size.

**Questions:**

- Could the author help clarify my confusion about the Feature decomposition strategy in section 5.2
- See weaknesses.

---

> ### Author Response · Authors · 2024-11-19
> **The responses to Reviewer KRan**
>
> We sincerely appreciate your valuable review comments and the effort you have put into them. We have addressed each of the questions you raised in detail. If anything is unclear, please inform us, and we will respond promptly.
>
> >*W1: Questions about Table 2.*
>
> **Response:** The feature decomposition strategy directly impacts the learning of seen and unseen species, as measured by the classification accuracy of old and new species presented in Table 2. To make the comparison more intuitive, we introduced a metric to quantify the similarity between "unique" and "common" features after decoupling. A lower similarity indicates a better decoupling effect. The experimental results are as follows, and RPL achieves the lowest similarity. We have updated this metric in the revised paper (Table 2).
>
> | Method    | Similarity |
> |-----------|------------|
> | CAM       | 6.53%       |
> | Grad-CAM  | 8.15%       |
> | RPL       | 1.89%      |
>
>
> >*W2: Issues about bolded accuracies.*
>
> **Response:** Thank you for your advice. In the revised paper, we have bolded only the highest performance results. As shown in Table 3, our method achieves the highest performance in the final session and exhibits the lowest performance drop (PD), indicating superior incremental learning ability.
>
> >*W3: Experiments should be run multiple times to ensure the robustness.*
>
> **Response:** In fact, for the CUB-200, PlantVillage, and Tree-of-Life datasets, we conducted 5 experiments with different random seeds and reported the averaged results to minimize the impact of randomness. We have included this information in the revised paper (Section 5.1).

---

> > ### Comment · Reviewer_KRan · 2024-11-25
> >
> > Thank you for clarifying Table 2 and the additional metric on feature similarity between 'unique' and 'common' features. Also, thank you for improving the readability of Table 3.
> >
> > Can you clarify if across your 5 different seeds, do you use the same 100 classes from the Tree-of-Life for each seed or are they a different set each time?

---

> > > ### Author Response · Authors · 2024-11-26
> > >
> > > Thank you for your response! We are pleased to provide further clarification on the question you raised, as outlined below:
> > >
> > > Our random sampling strategy for the Tree-of-Life dataset was inspired by the **Mini-ImageNet** dataset. The Mini-ImageNet dataset is widely used in regular few-shot class-incremental learning and consists of 100 randomly sampled classes from the original ImageNet dataset. Its purpose is to create a smaller, more manageable dataset while retaining the diverse data distribution and inter-class complexity of the original ImageNet. Similarly, we randomly sampled **13,487 images from 100 species across 42 families** in the Tree-of-Life dataset. This approach ensures both species diversity and the challenges arising from inter-species similarities within the same family.
> > >
> > > In our study, we conducted multiple experiments with different random seeds on the fixed set of 100 classes to enhance robustness and reproducibility. This detail has been updated in the revised paper (Section 5.1). This design offers two key advantages:
> > > 1. It minimizes the randomness of experimental results.
> > > 2. It provides a standardized benchmark, allowing future researchers to perform fairer performance comparisons.
> > >
> > > In fact, during our experimental design, we conducted comparative experiments using different randomly sampled class subsets from the Tree-of-Life dataset. Specifically, we randomly sampled six different groups of 100 classes and conducted experiments for each group. The results are as follows:
> > >
> > > | **Method** | **Session 0** | **Session 1** | **Session 2** | **Session 3** | **Session 4** | **Session 5** | **Session 6** | **Session 7** | **Session 8** | **PD** |
> > > |-------------|---------------|---------------|---------------|---------------|---------------|---------------|---------------|---------------|---------------|-------|
> > > | **Set 1**   | 50.32         | 47.25         | 43.12         | 41.56         | 38.29         | 37.76         | 35.16         | 35.03         | 33.02         | 17.30 |
> > > | **Set 2**   | 49.67         | 46.75         | 41.75         | 41.00         | 37.68         | 36.67         | 34.33         | 34.57         | 32.14         | 17.53 |
> > > | **Set 3**   | 50.21         | 48.35         | 43.81         | 42.98         | 39.29         | 38.22         | 35.98         | 36.25         | 34.02         | 16.19 |
> > > | **Set 4**   | 51.29         | 48.14         | 44.56         | 43.12         | 39.87         | 38.12         | 36.25         | 35.36         | 34.62         | 16.67 |
> > > | **Set 5**   | 49.89         | 46.93         | 42.06         | 41.55         | 38.02         | 36.99         | 34.78         | 35.01         | 32.54         | 17.35 |
> > > | **Set 6**   | 50.55         | 48.24         | 43.36         | 41.98         | 38.47         | 37.42         | 35.82         | 35.97         | 33.66         | 16.89 |
> > >
> > > From these results, we observe that the performance differences between different sets are not significant. Taking into account factors such as the number of species and families, we selected **Set 2** as the benchmark because it represents a more challenging case (its performance is the lowest among the six sets). And we will publicly release the class and data details for Set 2, enabling other researchers to build upon this benchmark for future studies.
> > >
> > > Thanks again for your invaluable suggestions! We sincerely hope that our response has addressed your concerns. Should you have any further questions or require additional clarification, please do not hesitate to contact us. We would be more than happy to engage in further discussion with you.

---

> ### Author Response · Authors · 2024-11-25
>
> Dear Reviewer KRan,
>
> We deeply appreciate the time and effort you have invested in evaluating our work. As the author-reviewer discussion period nears its conclusion, we kindly ask for your feedback on whether our responses have sufficiently addressed your concerns. If you have any additional suggestions or comments, please feel free to share them with us. We are more than willing to engage in further discussion and remain committed to making any necessary improvements.
>
> Once again, thank you for your thoughtful insights and invaluable suggestions. We look forward to your response!

---

> ### Comment · Reviewer_KRan · 2024-11-26
>
> Thank you for showing the generalizability across the tree-of-life dataset! I have updated my rating to accept, good paper.

---

> > ### Author Response · Authors · 2024-11-27
> >
> > We are truly grateful for your kind recognition of our work! It is our sincere hope that our efforts can meaningfully contribute to the development of the community.

---

### Official Review · Reviewer_P2DQ · 2024-11-04

**Soundness:** 2
**Presentation:** 2
**Contribution:** 2
**Rating:** 6
**Confidence:** 3

**Summary:**

The work observes that a limitation of incremental learning is confusion with closely related species (like species from same family) and therefore makes use of family label of each species to overcome that. The method learns one family level prototypes representing shared traits, and species level prototypes representing species specific traits. In the initial learning stage prototypes are learned through a method introduced called Residual Prototype Learning and during incremental learning stages a method called Residual Prototype Mixing is introduced.

**Strengths:**

1.	Extensive comparison with multiple relevant baselines for CUB200.
2.	Family information of any species is easy to obtain, so using that to improve incremental learning makes sense.

**Weaknesses:**

1.	Basis for some assumptions made are not clearly explained. Please refer to the questions.
2.	Not every baseline is considered for all three datasets. No particular reason is provided for skipping several baselines. Particularly I believe NC-FSCIL is a necessary baseline since its performance is pretty close to PA without having to use family label. If the skipped methods are not suitable for Tree-Of-Life and Plant-Village, please explain.
3.	No discussion of a possible limitation: In RPM are we assuming no new species is introduced whose family is not seen during the initial learning phase. If so, that is possible limitation of the work and it has to be discussed.

**Questions:**

1.	Line 274, how the assumption that prototype do not contain background information is made? Is it not possible that both family level and species level prototypes may contain background information.
2.	Line 275 and 276, how can it be assumed that the species level prototypes (mu_ck) contain both family-shared and species-unique information, while in the RPL (line 231 and 232) it is assumed they capture the most species unique traits.
3.	In section 4.3, it is not clear from Equation 3 how the delta of cosine similarity is optimized as mentioned in line 289. Is it happening implicitly when we optimize as in Equation 3.
4.	In section 5.3 Feature Distribution subsection, does “without PA” mean the baseline model from Table 1, if so, please mention it explicitly
5.	Throughout the main paper, I’m assuming the cosine similarity between three vectors as in Equation 2 and 3 means the summation (or average?) of cosine similarity of the feature vector with respect to the two prototypes, but it would be better to explicitly mention it.
6.	In table 3, PA’s performance is bolded. If the best performing method at each session can be bolded instead that would improve the readability of the table, making it easy to understand which method is doing better as I can see in Table 3, PA is not doing the best in all sessions (can also consider highlighting first, second and third best differently)

---

> ### Author Response · Authors · 2024-11-19
> **Part one of the responses**
>
> We sincerely appreciate your valuable review comments and the effort you have put into them. We have addressed each of the questions you raised in detail. If anything is unclear, please inform us, and we will respond promptly.
>
> >*W1: Missing several baselines on PlantVillage and Tree-of-Life datasets.*
>
> **Response:** Here are the reasons why not every baseline is applied to all datasets: In our experiments, we reproduced publicly available code and methods suitable for biological recognition applications on the PlantVillage and Tree-of-Life datasets. However, NC-FSCIL requires pre-calculating classifier heads for all categories, which means it assumes prior knowledge of the number of new categories that will appear. This constraint makes NC-FSCIL a closed-set classification algorithm, whereas in biological recognition scenarios, it is often impossible to predict the emergence of new categories. As a result, we did not apply this method to biological datasets in our primary evaluation.
>
> Nevertheless, we have tested the NC-FSCIL algorithm on the PlantVillage and Tree-of-Life datasets. The results are as follows:
>
> **PlantVillage**:
> | Method      | Session 0 | Session 1 | Session 2 | Session 3 | Session 4 | PD |
> |-------------|-----------|-----------|-----------|-----------|-----------|-----------|
> | NC-FSCIL    | 99.48     | **96.05**     | **93.59**     | 92.88     | 89.13     | 10.35     |
> | PA (ours)   | **99.50**     | 95.83     | 93.45     | **93.24**     | **90.51**     | **8.99**      |
>
> **Tree-of-Life**:
> | Method      | Session 0 | Session 1 | Session 2 | Session 3 | Session 4 | Session 5 | Session 6 | Session 7 | Session 8 | PD|
> |-------------|-----------|-----------|-----------|-----------|-----------|-----------|-----------|-----------|-----------|-----------|
> | NC-FSCIL    | **49.89**     | 46.52     | **42.08**     | **41.23**     | 37.56     | 36.49     | **34.45**     | 34.21     | **32.42**     | **17.47**     |
> | PA (ours)   | 49.67     | **46.75**     | 41.75     | 41.00     | **37.68**     | **36.67**     | 34.33     | **34.57**     | 32.14     | 17.53     |
>
> The results show that our method outperforms NC-FSCIL in the final session and achieves the smallest performance drop on the PlantVillage dataset, while delivering comparable performance (slightly lower than NC-FSCIL) on the Tree-of-Life dataset. Unlike NC-FSCIL, our method does not require prior knowledge of the total number of unknown classes, making it more scalable and practical. These results have been included in the revised paper (appendix).
>
>
> >*W2: Possible limitation of RPM?*
>
> **Response:** In fact, regardless of whether the new class's family appeared in the initial learning stage, RPM can work properly. Species confusion is more likely to occur when the new and old species belong to the same family. In contrast, when they do not share a family, the model only needs to perform standard few-shot incremental learning for the new species. Therefore, in Section 4.3, we focus on analyzing the case where the new and old species belong to the same family. Furthermore, when the family of a new class was not present during the initial learning stage, RPM generates additional feature samples by combining the new species features with the residual prototypes. This process helps mitigate overfitting caused by the limited number of training samples, as described in Lines 294–296 of the original paper.
>
> >*Q1: How is the assumption that prototypes do not contain background information made?*
>
> **Response:** The reasoning is as follows: As established in prior works, such as ProtoNet [1], learned prototypes represent the feature centers of their respective classes. Consequently, prototypes are expected to converge on category-specific features, while background features, being category-independent, are naturally excluded from the prototypes.
>
> [1] Prototypical Networks for Few-shot Learning, NeurIPS 2017.
>
> >*Q2: Why do species-level prototypes represent different meanings in the RPL and RPM stages?*
>
> **Response:** Here are the reasons: With sufficient training data for the base species, the model can learn more representative prototypes (Lines 231–232 of the original paper), and the "species-unique" and "family-shared" traits are further decoupled through the bi-directional optimization process defined in Eq. 2. However, during the incremental learning phase (Lines 275–276 of the original paper), the limited training data for new species makes it challenging for the model to learn accurate prototypes. As a result, the prototypes for these new species often contain coarse semantic information, incorporating both the most and second most discriminative features. This is why we employ RPM to guide the prototypes toward better convergence on species-unique features. We have emphasized this point in the revised paper (Section 4.3).

---

> > ### Comment · Reviewer_P2DQ · 2024-11-23
> >
> > Thanks to the authors for the responses.
> >
> > > *Q6: Issues about the bolded performance.*
> >
> > A minor comment, in Table 3 PD column I notice the best method appears to be FSLL. So please correct the bolding accordingly.
> >
> > > *Q1: How is the assumption that prototypes do not contain background information made?*
> >
> > With sufficient training the prototypes can converge towards the most species specific traits, but there can still be some background information that are also species specific. This is especially true in biological datasets, where some species can only be found in certain environmental conditions. Although, this can be mitigated with very large amount of data, that does not seem to be the case considering the experimental setting of this particular work.
> >
> > For example, let’s consider a scenario where there is only one species “A” in the set of base classes that lives in environment “E”. Since there is only one species, information about “E” will be part of $\mu_{c_i}$. Now if a new species “B” that also lives in “E” is introduced,
> > we don’t get the opportunity to augment the new species features with the information of “E” since it is not in the residual prototype $\mu_{r_i}$. This can lead to confusion between “A” and “B”.
> >
> > Essentially everything that is learned as specific unique need not truly be species unique.
> >
> > If this reasoning is valid, please consider highlighting this has a potential limitation. This is a corner case that is less likely to happen when there is sufficiently large amount of data.
> >
> > Apart from the above two points, I’m satisfied with the rest of responses.

---

> > > ### Author Response · Authors · 2024-11-24
> > > **The responses to Reviewer P2DQ**
> > >
> > > Thank you for your response and suggestions. Below, we provide further replies to Q6 and Q1:
> > >
> > > **Reply to Q6:**
> > > We have addressed this issue in the revised paper. While the PD of our method is slightly higher than FSLL by 0.12%, our final accuracy surpasses it significantly by 6.98%.
> > >
> > > **Reply to Q1:**
> > > If only one species is confined to a specific environment during the base training phase, it is possible for environmental features to be misidentified as species-specific traits. However, as long as the base training phase includes other species closely related to this species (which is the case in most scenarios), these environmental features will be learned as shared traits in the residual prototype (since closely related species are highly likely to inhabit similar environments), thereby reducing the risk of species confusion for new species in the same environment. Nevertheless, as you mentioned, to guarantee that the prototype contains no background information, it is crucial to ensure the diversity of the training data during the base training phase. We have discussed this limitation in the revised paper (Section 4.2, Lines 250–252).
> > >
> > > Thank you again for your valuable suggestions. We hope our responses can resolve your concerns.

---

> > > > ### Comment · Reviewer_P2DQ · 2024-11-24
> > > >
> > > > Thanks for the revisions. I've updated my rating to **6. marginally above the acceptance threshold**

---

> > > > > ### Author Response · Authors · 2024-11-24
> > > > > **The responses to Reviewer P2DQ**
> > > > >
> > > > > Thank you for recognizing our work and providing positive feedback! We look forward to the opportunity to share this work with the community.

---

> ### Author Response · Authors · 2024-11-19
> **Part two of the responses**
>
> >*Q3: Is the delta of cosine similarity optimized implicitly when we optimize as in Equation 3?*
>
> **Response:** Yes, this occurs during optimization as described in Eq. 3. Specifically, in Eq. 3, the cross-entropy loss function takes the labels and the cosine similarity scores between the features and all prototypes of $\boldsymbol{\theta}_c$ and $\boldsymbol{\theta}_r$ as input. To minimize the loss, the cosine similarity score for the target class must be the highest, which effectively maximizes the delta of cosine similarity.
>
>
> >*Q4: Does “without PA” mean the baseline model from Table 1?*
>
> **Response:** Yes, it refers to the baseline method in Table 1, and we have fixed this in the revised paper.
>
>
> >*Q5: Questions about the calculation of cosine similarity.*
>
> **Response:** Actually, the cosine similarity is computed by calculating the similarity between the feature and each prototype individually, resulting in a set of scores that serves as the standard output of a classification model. We have clarified this point in the revised paper (Section 4.2) to prevent any potential misunderstandings.
>
> >*Q6: Issues about the bolded performance.*
>
> **Response:** Thanks for your advice. We have bolded only the highest performance results in the revised paper to fix this issue.

---

### Official Review · Reviewer_qswS · 2024-11-05

**Soundness:** 1
**Presentation:** 1
**Contribution:** 2
**Rating:** 3
**Confidence:** 4

**Summary:**

This paper studies "fine-grained" class-incremental learning in a few-shot setting. The authors argue that under such a setting, the model would easily suffer overfitting (to new classes) or forgetting (of old classes). They thus propose to leverage hierarchical-class information (e.g., family-species) to encourage knowledge sharing across species of the same family while learning discriminative information. The paper proposes two new learning strategies for this purpose: RPL and RPM. In three small-scale experiments, the proposed approach demonstrated improved performance.

**Strengths:**

S1. The paper focuses on an interesting and challenging problem.

S2. The paper points out several insights that future research on fine-grained incremental learning could leverage.

**Weaknesses:**

W1. It is unclear or not intuitive why when new classes and old classes are similar (so have shearable information), incremental learning will lead to forgetting.

W2. The technical/implementation details are not clear and not optimally designed. First, ResNet18 seems to be a too-weak backbone, especially for fine-grained problems. The authors may consider transformers (DINO, DINO-v2, BioCLIP visual encoder) or ResNet50 at least. Second, there is no information about whether the backbone is pre-trained, and if so, on what dataset. Third, it seems the feature backbone is only updated during the base training stage (Eq 1). If so, suppose the linear classifiers of the base (old) classes are frozen during continual learning, I see no reason why the old classes will be forgotten. Fourth, do the authors impose an orthogonal constraint in Line 270?

W3. My major concern is the approach itself. The design seems to be quite ad hoc without justifications. First, I'm not sure why computing the residual between the linear classifiers (or prototypes) and feature vectors makes sense. Please note that the prototypes can simply be the feature vectors times a scalar. Why does the residual contain information on "the secondary discriminative features likely representing traits shared across the family?" Second, why does adding the residual to feature vectors of the new species make sense? What do the new feature vectors encode? Third, the meanings of the decompositions and terms introduced in Lines 270 - 290 are not clear or justified. Finally, if the residual is added to features of the new species during training, how about the inference stage?

W4. The experiment details are missing; the experimental design can be improved. For example, no dataset statistics are provided. No information about how the authors obtained the family information. The PlantVillage dataset seems to be too small; the authors could consider using families in iNaturalist. No information about how the Tree of Life Dataset (with ~400K species) is subsampled. How do the authors ensure that all the families are seen during the base training time? Finally, experiments on no more than 200 species are a bit too small.

=== Minor ===

W5. There are many missing references, for example, no references to the datasets in the introduction; no reference to "Inspired by human’s reference-based learning mechanism."

W6. The related work seems to miss one topic, generalized few-shot learning, such as "Generalized zero-and few-shot learning via aligned variational autoencoders, CVPR 2019." Will this line of work resolve the problem in Line 138?

**Questions:**

Q1. What is the detail of the Feature decomposition strategy (Line 400)? CAM and Grad-CAM are methods to visualize a classifier. How did the authors use their saliency responses to get accuracy in Table 2?

Q2. BioCLIP is a foundation model pre-trained on over 400K species. Can the authors provide more details about the argument that "it struggles with continual learning and tend to misclassify closely related species?" (Line 126)

---

> ### Author Response · Authors · 2024-11-19
> **Part one of the responses**
>
> We sincerely appreciate your valuable review comments and the effort you have put into them. We have addressed each of the questions you raised in detail. If anything is unclear, please inform us, and we will respond promptly.
>
> >*W1: Why species similarity leads to forgetting.*
>
> **Response:** The forgetting is caused by species confusion. As noted in lines 15–16, "... often faces species confusion, leading to 'catastrophic forgetting' of old species...", Fig. 1 provides a detailed illustration of this phenomenon. The confusion between old and new species during incremental learning results in a significant drop in the performance of old species, a process known as 'old class forgetting'.
>
> >*W2-first: choice of backbone.*
>
> **Response:** We address this question from three key aspects:
> 1. As stated in Line 353, we selected ResNet18 as the backbone to "maintain consistency with prior works." ResNet18 is the standard backbone adopted by most mainstream and state-of-the-art methods for few-shot incremental learning [1][2][3][4][5], ensuring fair comparisons and alignment with established benchmarks.
> 2. While "large foundation models" exhibit strong representational capabilities and have been widely applied across various scenarios, their use in few-shot incremental learning introduces the issue of data leakage. These models are pretrained on extensive datasets that often overlap significantly with the datasets used in mainstream few-shot incremental learning benchmarks. For instance, categories such as animals and plants commonly appear in the pretraining datasets of models like DINO and BioCLIP. As a result, the "new classes" in incremental learning are not genuinely unseen, which undermines the objectivity and validity of algorithm evaluations.
> 3. Our primary contribution, the prototype antithesis strategy, is not confined to ResNet18 and can be seamlessly integrated with other backbones, including ResNet-50, ResNet-101, and transformer-based architectures. To demonstrate its generalizability, we conducted experiments on the CUB200 dataset using ResNet-50, with the results showing below, from which we can see that PA (ours) further improves against the baseline method with significant margins.
>
> | Method             | Session 0 | Session 1 | Session 2 | Session 3 | Session 4 | Session 5 | Session 6 | Session 7 | Session 8 | Session 9 | Session 10 |PD |
> |--------------------|-----------|-----------|-----------|-----------|-----------|-----------|-----------|-----------|-----------|-----------|------------|------------|
> | Baseline (ResNet-50) | **83.44**     | 79.14     | 75.81     | 71.08     | 69.97     | 67.16     | 66.25     | 65.01     | 62.68     | 62.84     | 62.08 |  21.36 |
> | PA (ResNet-50)      | 83.07     | **80.44** | **76.91** | **72.17** | **72.21** | **69.92** | **68.63** | **67.05** | **66.04** | **65.12** | **64.75**  |   **18.32** |
>
> >*W2-second: Is the backbone pretrained?*
>
> **Response:** Following prior works [1][2][3][4][5], we utilize ResNet-18 pretrained on ImageNet for experiments on the CUB200 dataset, while a randomly initialized ResNet-18 is used for the PlantVillage and Tree-of-Life datasets. We have included this information in the revised paper (Section 5.1).
>
> >*W2-third: Why old classes will be forgotten?*
>
> **Response:** The old classes could be forgotten even if the backbone and old class prototypes are fixed, here are the reasons: Even with the feature backbone and old class classifiers frozen, newly added class classifiers can disrupt the classification probability distribution, leading to misclassifications and performance degradation of old classes, termed as "old class forgetting". For example, when a new class is highly similar to an old class, both old and new classifiers may produce high classification scores, increasing the likelihood of misclassifying an old class as a new one. This issue is also discussed in CEC [2] (Introduction, Paragraph 3), where both the feature backbone and old class classifiers are frozen during incremental learning.
>
> >*W2-fourth: Do the authors impose an orthogonal constraint in Line 270?*
>
> **Response:** It is not an imposed constraint, any feature vector can inherently be decomposed into two orthogonal components due to the properties of vector spaces. Specifically, given a vector **a** and an auxiliary vector **b**, **a** can be expressed as:
>
> 1. A component parallel to **b**:
>    **u** = $Projection_{to-b}$(**a**),
> 2. A component orthogonal to **b**:
>    **v** = **a** - **u**.
>
> These components satisfy **a** = **u** + **v** and **u** $\cdot$ **v** = 0. The existence of projections guarantees that this decomposition is always possible. For ResNet-18, the feature vectors for each image, which reside in $\mathbb{R}^{512}$, can similarly be orthogonally decomposed along a species-unique direction.

---

> ### Author Response · Authors · 2024-11-19
> **Part two of the responses**
>
> >*W3-first: Why computing the residual between the prototypes and feature vectors makes sense? And why residuals represent family traits?*
>
> **Response:** The computation of the residual is well-justified. As demonstrated in prior few-shot learning works like ProtoNet [6], prototypes naturally converge to the centers of their respective feature distributions. In our approach, prototypes are used as classifiers by directly calculating the feature distances between prototypes and features without scaling (refer to Lines 207–208 of the original paper). This alignment ensures that both features and prototypes reside in the same space, making the residual operation meaningful. Additionally, GAN-based methods, such as AGE [7], also adopt this approach to remove category-related attributes effectively.
>
> Notably, during the bi-directional optimization defined in Eq. 2 under "family-level" label supervision, residual prototypes—not residual features—converge to "secondary discriminative features," representing family-shared traits.
>
> >*W3-second: Why adding the residual to the new species features make sense?  What do the new feature vectors encode?*
>
> **Response:** This is because both the residual and new species features are encoded from input images by the same feature extractor. Since they exist in the same aligned feature space, adding the residual to the feature vectors of the new species represents a linear feature fusion strategy. And the newly obtained feature vectors represent the new species features enhanced with family-shared traits (as described in Lines 271-274 of the original paper).
>
> >*W3-third: The meanings of the decompositions and terms introduced in Lines 270 - 290.*
>
> **Response:** Lines 270-290 argue why RPM helps the model focus more on "species-unique" features in new categories, based on the classification model optimization process. To simplify, we decompose both features and prototypes into species-related and species-unrelated parts.
> For features, the species-unrelated part is further divided into family-shared and background components, while for prototypes, the species-unrelated part is simply the family-shared component.
> When the new and old species belong to the same family, RPM enhances the family-shared part of the new species by mixing the old species' family-shared features with the new species' features. This makes the classifier likely to produce a high response to the family-shared part, potentially causing the model to misclassify it as the family label.
> To avoid this, the model enhances the response of the species-unique components. This is achieved through the optimization of Eq. 3, as mathematically detailed in Lines 285–291 of the original paper.
>
> In the revised paper, we have polished this section to make it more intuitive and easier to understand.
>
> >*W3-fourth: Details about the inference stage.*
>
> **Response:** During inference stage, the features of the new species are classified by the prototypes without mixing with the residual. This is because, through RPM, the model has established well-separated decision boundaries, allowing it to achieve high accuracy when directly classifying new categories. And we have added this information in the revised paper (Section 4.1).

---

> ### Author Response · Authors · 2024-11-19
> **Part three of the responses**
>
> >*W4: Questions about the experiment details.*
>
> **Response:**
> 1. Dataset statistics: In the Experiment Settings section, we focus on describing the categories of the dataset and the way it is partitioned, and the family information are directly obtained from their classification labels. Additional statistics, including the number of images, species, and families, are provided in the revised paper (appendix).
>
> 2. Although the PlantVillage dataset contains a relatively small number of categories, it is a classification dataset focused on leaf diseases caused by pests and pathogens, which holds significant importance in biological research. Furthermore, as the dataset consists of various lesions on different types of leaves, the similarity between categories is very high, making it prone to category confusion during incremental learning. This makes it a highly challenging dataset.
>
> 3. The Tree-of-Life and iNaturalist datasets have a significant overlap, which is why we did not choose the iNaturalist dataset as the benchmark for this paper. Nonetheless, we randomly sampled 500 species from the iNaturalist 2021 dataset for exploratory experiments (using ResNet-50 as the backbone). These 500 species span 136 families and contain a total of 103,582 images. Among them, 340 species were designated as base species, while the remaining 160 species were evenly divided into 8 incremental stages, with 20 new species added per stage. The experimental results (shown below) demonstrate that, compared to the baseline, our method maintains a clear performance advantage in incremental learning on the iNaturalist 2021 dataset.
>
> | Method   | Session 0 | Session 1 | Session 2 | Session 3 | Session 4 | Session 5 | Session 6 | Session 7 | Session 8 | PD|
> |----------|-----------|-----------|-----------|-----------|-----------|-----------|-----------|-----------|-----------|-----------|
> | Baseline | 54.38     | 51.23 | 45.91 | 45.13     | 41.21 | 40.82 | 38.67     | 38.31 | 36.16     | 18.22     |
> | PA (ours)      | **57.65**     | **54.41** | **48.55** | **48.02**     | **44.28** | **43.68** | **41.29**     | **41.08** | **39.76**     | **17.89**     |
>
>
>
> 4. As described in Line 350, the 100 species are randomly sampled from the Tree-of-Life dataset, and we have the included the detailed statistics in the revised paper (appendix).
>
> 5. We do not need to ensure that all families appear during the base training phase. If the family of a new class does not appear during the base training phase, it indicates that the new class differs to some extent from the old classes, and in this case, we can directly proceed with classification. Our method is designed to handle scenarios where new and old classes are extremely similar, which is clearly a more challenging situation.
>
> 6. The mainstream datasets for few-shot incremental learning typically have fewer than (or equal to) 200 categories. For example, CUB-200 has 200 categories, mini-ImageNet has 100 categories, and CIFAR100 has 100 categories. The challenge of this task lies in the forgetting and overfitting that occurs during the incremental learning process. As the amount of data decreases in each incremental stage, the model's learning becomes more difficult. Especially in biological recognition scenarios, the number of new or mutated categories and samples is often very small, so the setup of this paper is closer to real-world application scenarios. However, we appreciate your suggestion, and we will explore experiments in scenarios with more categories in the future.
>
> >*W5 and W6: Missing reference.*
>
> **Response:** We cited all datasets in Lines 342–343 of the original paper. To enhance clarity, we have also cited them in the introduction of the revised paper. Additionally, we have included a citation for "Inspired by human’s reference-based learning mechanism" [8] in the revised paper.
> We have also included the discussion about GFSL [9] in the revised paper (section 2):
> "GFSL addresses this (forgetting) by aligning old and new class distributions to mitigate forgetting, but it struggles when old and new classes distribution gaps are large, and risks class confusion when old and new classes are highly similar".

---

> ### Author Response · Authors · 2024-11-19
> **Part four of the responses**
>
> >*Q1: Questions about Table 2.*
>
> **Response:** The feature decomposition strategies include RPL, CAM and Grad-CAM. The details are stated in Lines 402–404 of the original paper: "Specifically, we set a threshold to extract the highest-response regions from the feature maps as unique features, while the secondary-high-response regions represent common features." The feature decomposition strategy plays a crucial role in the learning process for seen and unseen species, as reflected in the classification accuracy reported in Table 2.
>
> To make the comparison more intuitive, we introduced a new metric to quantify the similarity between "unique" and "common" features after decoupling. A lower similarity score indicates a more effective decoupling process. The experimental results (shown below) demonstrate that RPL achieves the lowest similarity score, highlighting its superior performance. This similarity metric is updated in the revised version of Table 2.
>
> | Method    | Similarity |
> |-----------|------------|
> | CAM       | 6.53%       |
> | Grad-CAM  | 8.15%       |
> | RPL       | 1.89%      |
>
> >*Q2: Questions about BioCLIP.*
>
> **Response:** According to the official data provided by BioCLIP, despite being pre-trained on a large number of animal and plant datasets, BioCLIP's accuracy on the PlantVillage dataset remains low in both zero-shot and 5-shot settings, achieving accuracies of 24.4\% and 80.9\%, respectively. These results are significantly below the average performance of PA's incremental learning (94.51\%), suggesting that BioCLIP struggles with substantial class confusion when dealing with closely related species, such as leaf lesions.
>
> Additionally, we conducted continual learning experiments using the official BioCLIP source code on both the CUB200 and PlantVillage datasets. The results demonstrate a notable decline in BioCLIP's average classification performance, as detailed below.
>
> **PlantVillage:**
> | Method     | Session 0 | Session 1 | Session 2 | Session 3 | Session 4 | PD |
> |------------|-----------|-----------|-----------|-----------|-----------|-----------|
> | BioCLIP    | **99.86**     | 60.75     | 59.65     | 54.54     | 52.57     | 46.89     |
> | PA (ours)  | 99.50     | **95.83**     | **93.45**     | **93.24**     | **90.51**     | **8.99**      |
>
> **CUB200:**
> | Method         | Session 0 | Session 1 | Session 2 | Session 3 | Session 4 | Session 5 | Session 6 | Session 7 | Session 8 | Session 9 | Session 10 | PD |
> |----------------|-----------|-----------|-----------|-----------|-----------|-----------|-----------|-----------|-----------|-----------|------------|------------|
> | BioCLIP [1]    | **79.23**     | 71.77     | 69.15     | 63.44     | 61.72     | 59.92     | 57.32     | 54.89     | 52.41     | 50.16     | 49.57      | 29.66      |
> | PA (ours)      | 78.69     | **75.59** | **72.71** | **68.71** | **68.37** | **65.77** | **64.75** | **63.59** | **62.76** | **62.02** | **61.19**  | **17.50**  |
>
> **Reference**
>
> [1]  Fewshot class-incremental learning, CVPR 2020.
>
> [2] Few-shot incremental learning with continually evolved classifiers, CVPR 2021.
>
> [3] Few-shot class-incremental learning via class-aware bilateral distillation, CVPR 2023.
>
> [4] Few-shot classincremental learning via training-free prototype calibration, NeurIPS 2023.
>
> [5] Orco: Towards better generalization via orthogonality and contrast for few-shot class-incremental learning, CVPR 2024.
>
> [6] Prototypical Networks for Few-shot Learning, NeurIPS 2017.
>
> [7] Attribute Group Editing for Reliable Few-shot Image Generation, CVPR 2022.
>
> [8] Cognitive computational neuroscience, Nature neuroscience, 2018.
>
> [9] Generalized zero-and few-shot learning via aligned variational autoencoders, CVPR 2019.

---

> ### Author Response · Authors · 2024-11-25
>
> Dear Reviewer qswS,
>
> We deeply appreciate the time and effort you have invested in evaluating our work. As the author-reviewer discussion period nears its conclusion, we kindly ask for your feedback on whether our responses have sufficiently addressed your concerns. If you have any additional suggestions or comments, please feel free to share them with us. We are more than willing to engage in further discussion and remain committed to making any necessary improvements.
>
> Once again, thank you for your thoughtful insights and invaluable suggestions. We look forward to your response!

---

> > ### Comment · Reviewer_qswS · 2024-12-03
> > **Response by the reviewer**
> >
> > Dear authors,
> >
> > I truly appreciate your additional explanations and results.
> >
> > They resolve parts of my concerns, but not all. I understand that there is limited time for the authors to provide further responses, so I will only ask for clarifications, not further experiments.
> >
> > W1: Why species similarity leads to forgetting.
> > Can the authors discuss further the forgetting led by similar classes and dissimilar classes? In my humble opinion, the latter should lead to more significant forgetting. This is why I get confused.
> >
> > W2-first: choice of backbone.
> > I understand the concern about other backbones. However, backbones like DINO do not use labeled information at all, can the authors further discuss the feasibility of using it? Further, it seems that the gain on ResNet-50 is small (maybe I'm wrong). How about other baselines on top of ResNet-50?
> >
> > W3:
> > This is still my main concern. I appreciate the authors for the detailed explanations, but I feel the paper requires major revisions to make the method part more accessible.

---

> ### Author Response · Authors · 2024-12-04
> **The responses to Reviewer qswS**
>
> Thank you for your reply. We would like to provide further clarification on the concerns you raised:
>
> **Response to W1**:
> We analyze the two scenarios you mentioned separately:
> 1. When the new classes are significantly different from the old ones, the parameter drift caused by model updates during the learning of new classes results in a performance drop for old ones, leading to "forgetting". This issue has been the focus of most prior studies and can be mitigated by freezing network parameters.
> 2. However, when the new classes are highly similar to the old ones, the classifier tends to confuse them. For instance, the model might misclassify an old class as a similar new class, as we explain the reasons and mechanisms behind this phenomenon in Figure 1 of our paper. This also causes a drop in the classification accuracy of the old classes, leading to "forgetting". Such a scenario, where new and old classes are similar, is common in biological recognition tasks and remains a significant challenge that has not been well-addressed in the few-shot incremental learning domain. In this paper, we specifically focus on addressing this issue.
>
> **Response to W2**:
> Feasibility of using DINO: DINO adopts a self-supervised pretraining approach, which indeed does not require explicit labels. However, during the pretraining phase, it is exposed to a large number of categories and learns their corresponding data distributions. Many of these categories are likely to appear in few-shot learning benchmarks, such as various plants and animals. This exposure could influence the model's incremental learning process; for instance, the model might leverage the learned data distributions to classify these categories by measuring the similarity between distributions. Consequently, from a rigorous academic perspective, this scenario constitutes data leakage, as the new categories are not genuinely unseen by the model. Nonetheless, if practical applications take precedence over strict academic rigor, DINO could be considered a viable backbone.
>
> Gain on ResNet-50: With ResNet50 as the backbone, our method achieves an improvement of 2.67% in the final classification accuracy across all species and a reduction of 3.04% in 'PD' compared to the baseline, which is a considerable improvement on the CUB200 dataset. For example, in current SOTA algorithms like NC-FSCIL [a] and TEEN [b], the final classification accuracy differs by merely 0.13%. Due to time constraints, we only conducted additional experiments using NC-FSCIL [a] with ResNet50 as the backbone for comparison. The results demonstrate that our method achieves a notable improvement of 1.19% in final classification accuracy (64.75% vs. 63.56%) and a reduction of 1.37% in 'PD' (18.32% vs. 19.69%) compared to NC-FSCIL [a]. This further validates that our method offers consistent advantages across different backbones.
>
> **Response to W3**:
> If you could kindly share your concerns regarding the description of our method in more detail and with greater clarity, it would greatly assist us in addressing your questions more effectively. In addition, we would like to outline the logical structure of the methodology section in our paper for your reference:
>
> 1. We first introduce the overall optimization process of the model. (Section 4.1)
> 2. Next, we detail the step-by-step optimization objectives and technical implementation details, progressively refining the description. (Section 4.2 and 4.3)
> 3. Finally, we provide theoretical justification for the rationale and effectiveness of our method. (Section 4.4)
>
> We hope our responses adequately address your concerns. If so, we would sincerely appreciate your consideration in revising the scores accordingly. Thank you for your time!
>
> **References**
>
> [a] Neural collapse inspired feature-classifier alignment for few-shot class-incremental learning.
>
> [b] Few-shot class-incremental learning via training-free prototype calibration.

---

### Author Response · Authors · 2024-11-19
**General Response**

We thank all reviewers for their time and positive evaluation of our work. Namely, our paper addresses "an interesting and challenging problem" (qswS) and introduces "novel training methods that improve class separating and alleviate catastrophic forgetting"(KRan). Our proposed method "makes sense to improve incremental learning"(P2DQ) and demonstrates "the generalization of their ability by evaluating on three biological image datasets"(KRan). Furthermore, our paper provides "a theoretical analysis of their method"(KRan) and includes "a detailed and comprehensive literature review"(Zao3), along with "a suite of experiments that is admittedly comprehensive"(Zao3).

We also sincerely thank all the reviewers for their insightful comments and suggestions. We have uploaded a revised version of the paper, incorporating the reviewers' helpful suggestions. The main changes are listed below (detailed in the individual responses):
- Expanded references and additional discussion of related works.
- Clarification of the inference stage process in detail.
- Detailed explanation of the cosine similarity calculations in Eq. 1, Eq. 2, and Eq. 3.
- Explanation of the differences between species prototypes learned during base training and incremental learning.
- Additional experimental details, including whether the backbone is pretrained, robustness of results, and more dataset statistics.
- Improvements to Fig. 3, Table 2, and Table 3 to enhance clarity and readability.
- Inclusion of comparisons with more prior works, such as NC-FSCIL, on the PlantVillage and Tree-of-Life datasets.

To facilitate the review process, we have marked the revised parts in the paper in red and provided point-by-point responses to each reviewer's comments. We hope this revised version is clearer, more complete, and demonstrates improved theoretical justification and empirical validation.

Once again, we extend our gratitude to the reviewers for their efforts and valuable feedback, which have greatly contributed to enhancing the quality of this paper.

---

### Author Response · Authors · 2024-11-27
**Kindly Request for Final Reviewer Queries and Score Adjustments**

We extend our heartfelt gratitude to the reviewers for their insightful comments and invaluable feedback, which have greatly contributed to improving our work. As the discussion period draws to a close, we warmly encourage any additional questions or requests for clarification. Should our responses have sufficiently addressed your concerns, we would sincerely appreciate your consideration in revising the scores and confidence levels accordingly. Thank you for your time, effort, and thoughtful engagement throughout this review process.

---

### Meta-Review · Area_Chair_Fh7n · 2024-12-19

**Metareview:**

This work studies the problem of few-shot class-incremental learning in the setting of biological species recognition. To solve the "catastrophic forgetting" problem, the authors designed a Prototype Antithesis method, which reduced the confusion between old and emerging species leveraging the hierarchical structures in biological taxa. This idea is interesting and novel. Besides, the experiments are extensive; on three large datasets the proposed method significantly reduced the inter-species confusion and achieved SOTA. In addition, there are still some minor issues, for example, lacking more explanations of the motivation and method design, which could still be enhanced. Overall, this work could be accepted.

**Additional Comments On Reviewer Discussion:**

I appreciate the authors work on the rebuttal and try to answer some detailed concerns by the reviewers. Some reviewers gain better understanding of the method and raise their scores. Although some reviewers still have some concerns regarding the motivation (i.e., the forgetting problem) and method design, the authors' rebuttal present more details. The motivation and idea make sense to me. Moreover, I think the authors' response to Reviewer qswS are clear, at least addressing most of his/her concerns (though Reviewer qswS hasn't raised his/her score). In addition, other reviewers are positive towards this work.

---

### Decision · Program_Chairs · 2025-01-22

Accept (Poster)